# Exposure to Respirable Particulate Matter and Its Association with Respiratory Outcomes in Beauty Salon Personnel

**DOI:** 10.3390/ijerph20032429

**Published:** 2023-01-30

**Authors:** Denis Vinnikov, Zhanna Romanova, Aizhan Raushanova, Arailym Beisbekova, Ermanno Vitale, Gulnar Bimuratova, Venerando Rapisarda

**Affiliations:** 1Environmental Health Laboratory, al-Farabi Kazakh National University, Almaty 050040, Kazakhstan; 2Occupational Health Risks Laboratory, Peoples’ Friendship University of Russia (RUDN University), Moscow 117198, Russia; 3Department of Nutrition, Asfendiyarov Kazakh National Medical University, Almaty 050012, Kazakhstan; 4Department of Clinical and Experimental Medicine, Occupational Medicine, University of Catania, 95124 Catania, Italy; 5City Polyclinic #7 of the Public Health Department of Almaty, Almaty 050040, Kazakhstan

**Keywords:** respirable, PM, occupational, exposure, beauty salon

## Abstract

We aimed to assess exposure to respirable particulate matter (PM) of beauty salon personnel, identify its determinants and ascertain the associated respiratory effects. We collected 122 full-day respirable PM samples from 12 beauty salons (floor area ranging from 24 to 550 m^3^, staff from 4 to 8) in Almaty, Kazakhstan, taking 10 samples from each place using a portable SidePak AM520 monitor. We also assessed lifestyle (smoking, etc.), respiratory symptoms and health-related quality of life (HRQL) of the personnel using questionnaires. Out of 11,831 5-min data points, daily median respirable PM concentrations were highly variable and ranged from 0.013 to 0.666 mg/m^3^ with 8.5-times difference in the median concentrations between the venue with the highest median (0.29 mg/m^3^) and the least median (0.034 mg/m^3^). In a multivariate linear regression modelling, ambient PM_2.5_ concentration was the strongest predictor of daily median respirable PM concentration (beta 2.12; 95% CI 1.89; 2.39), and R^2^ of the model was 0.63. We also found a positive association of the median respirable PM with respiratory symptoms and seasonal allergy, but not with HRQL. Short-term respirable PM levels in the beauty salons may be very high, but the median concentrations are mainly determined by the ambient air pollution.

## 1. Introduction

Occupational exposure to vapors, gases, dusts and fumes via the inhalational route is frequent in a wide range of jobs. Many respiratory diseases, including chronic obstructive pulmonary disease (COPD) are associated with such occupational exposure [1]; however, the associated health effects are not limited to respiratory disease and extend beyond chronic conditions. Depending on the chemical composition of these substances, the risk of selected malignancies may also be elevated [2,3,4]. Work in the beauty industry may represent one of the occupations with high risk due to frequent use of chemicals, including formaldehyde [5], substances with sensitizing effect [6], ammonia [7], volatile organic compounds (VOC) [8], bleachers, dyes and other substances. VOCs encompass a wide range of compounds including, but not limited to, aromatics (toluene, xylene), esters and ketones (ethyl acetate, acetone) and terpenes (pinene, limonene, camphor, menthenol), the effects of which have yet to be fully understood. Of note, knowledge on hazards associated with employment in the beauty industry among workers may not be sufficient, given that the practices may not always be safe [9].

Studies which reveal personal exposure of beauty salon employees to chemicals and particulate matter (PM), generated from various sources, mostly from sprays, are limited. Overexposure to VOCs [8], formaldehyde [5] and other irritants, sensitizers and other hazardous chemicals entails respiratory health effects, even forcing hairdressers to change jobs quite often [10]. At present, there is no clear understanding on the levels of personal exposure and predictors of such exposure, although several studies have concluded that beauty salon employees are often exposed to a complex mixture of chemicals, and that quantification of health effects will be challenging. The effects of air conditioning, window opening, air moisturization as well as the confounding effects of available space per worker, personal lifestyle/smoking/exposure to secondhand smoke need to be further evaluated in larger studies. Even when protective measures are in place, they still do not preclude high exposure to PM, as respirable PM concentration may rise up to very high levels [11]. Furthermore, in places with severe ambient air pollution, high indoor concentrations will likely have an additive effect on respiratory health and other systemic health outcomes.

Concerns over the health effects of occupational exposure of a hairdresser mostly arise from the use of sprays, which produce aerosols, usually a mixture of many chemicals. The respirable fraction of such aerosola may be of greatest interest because of the associated health effects, demonstrated in a wide range of occupations. While there are a few existing reports on the indoor respirable PM monitoring in the beauty salons, and they show peaks of very high mass concentrations, these studies were limited to a few geographical areas and brands of sprays, solvents and other substances used locally [11] in addition to a limited number of days of monitoring. Moreover, we found no studies highlighting the association of exposure levels with respiratory outcomes and clear quantification of risk. We, therefore, aimed to assess exposure to respirable PM of beauty salons’ personnel, to identify its determinants and ascertain the associated respiratory effects.

## 2. Materials and Methods

This study was approved by the Committee on Bioethics of al-Farabi Kazakh National University and was conducted in accordance with the Declaration of Helsinki and all relevant regulations. All study participants signed a written informed consent to participate. For respirable PM monitoring we selected twelve locations in Almaty, the largest city of Kazakhstan, located all over the city and stretching from the northern (also lower altitude) to the southern locations (closer to the hills). Selection of locations was dictated mostly by salons’ consent to participate, but we also pursued the need to embrace neighborhoods differing in presumed air pollution, population socioeconomic status and population density. Aerosol monitoring was completed within 10-day periods in each place in order to cover business days and weekends, and to show maximum variation in concentrations resulting from alternating workloads. All the included salons provided service to both males and females and did not have any specific specialization. In addition to respirable PM monitoring, we also asked available personnel to complete the questionnaire.

The questionnaire was offered in either Russian or Kazakh to all those willing to participate and consisted of 58 questions. We found no differences in age and sex between the groups of those who agreed and refused to participate. We have reported earlier the outcomes of a population-based study ascertaining occupational profile and the risk of respiratory disease, in which we applied this questionnaire [12]. In brief, detailed demographic information, occupational history, including current occupation and work duration for each position held in a lifetime, socioeconomic information, cigarette, waterpipe and electronic cigarette smoking, exposure to secondhand smoke at work or at home, use of fossil fuel at home for cooking or heating, physical activity and alcohol use comprised the first part of this questionnaire. We then offered two standardized tools to assess current respiratory symptoms, including CAT and mMRC. Furthermore, any history of wheezing, allergy and doctor diagnosed illness was included in this part. A short version of the health-related quality of life (HRQL) tool, SF-8, and a detailed occupational exposure questionnaire including in the past constituted the final part of this questionnaire.

Because selected salons did not have a fixed work schedule and could operate late into the day and we sampled respirable PM mass concentrations in these places from the time of their opening till their closure, sampling duration differed from day to day. We used a portable, battery-operated monitor SidePak AM520 (TSI, USA), which was placed at an elevation of 1.5–1.7 m from the floor in the central part of the main hall. This device used light-scattering photometric technology to measure PM mass concentration. A hose attached to the inlet of the monitor was connected to a Dorr-Olivier cyclone, set to operate at a flow rate of 1.7 L/min. The device was zeroed daily prior to sampling. There was only one device in each sampled venue, and we collected ten full-day samples corresponding to ten consecutive days of monitoring in each place. The monitor was set to log 5-min means of respirable PM, which were treated as data points in our analysis. For a typical 8-h workday, we collected 96 data points of respirable PM mass concentrations. In the subsequent analyses, we computed daily medians from these 5-min points, expressed as mg/m^3^, but we also presented the minimum and maximum (peak) concentrations for each day.

In this study, we did not aim to monitor ambient air concentrations of respirable PM; instead, we used sampling data from a network of portable sensors, integrated into the State Agency on environmental monitoring (Kazhydromet). Data on the daily mean concentrations of PM2.5 were available from the website www.airkaz.org (accessed on 9 January 2023). This resource pools 1-min data from portable sensors situated throughout the city to daily means, from a number of cities in Kazakhstan. We extracted data from all sensors in Almaty (nine to eleven active sensors, depending on the day) for each day when we collected respirable PM concentrations in the salons, and calculated the means for those days. Because respirable PM concentrations are not measured by these sensors, we only had PM2.5 concentrations available for this analysis.

The primary outcome of this analysis was daily the median respirable PM mass concentration in mg/m^3^. This concentration was computed from 5-min mean concentrations as data points. First, we tested normality of all data with Shapiro–Wilk test and found that respirable PM mass concentrations along with most other continuous data in the current presentation were non-normally distributed. Therefore, all reported tests in our presentation were nonparametric. For respirable PM concentrations, we present minimum, maximum and median concentrations for each day of sampling. We then compared daily medians (122 days) with each other using the Kruskall–Wallis test. Furthermore, median and geometric mean concentrations were calculated for each of the 12 included salons from daily means in each place (from 10 data points representing daily means). These medians and geometric means for each salon were tested in a Kruskall–Wallis test to assess whether the variance between salons (between-group) exceeded the one within them (within-groups).

In order to identify predictors of the log-transformed daily indoor median respirable PM concentrations, we applied linear regression modelling, in which we included log-transformed ambient PM2.5 concentrations for a given day, internal square, the mean number of customers a day, weekday (business day vs. weekend) and number of working personnel and reported beta coefficient for these predictors along with their corresponding 95% confidence intervals (CI). None of these predictors were collinear.

Demographic, lifestyle, occupational history, respiratory symptoms and HRQ were described and analyzed assuming normal distribution and thus presented as means with the corresponding standard deviation (SD) in those few cases when data were normally distributed; or, alternatively, they were presented as medians with their corresponding interquartile range (IQR). Binary data distributions are presented as N with their percent from the group, and, whenever applicable, χ^2^ test was used to test difference between two groups. HRQL mean scores were compared to the population means for Almaty [13] using the Mann–Whitney test. We then aimed to see whether indoor respirable PM concentrations differed in the univariate comparisons of respiratory symptoms, respiratory diagnoses or HRQL; for that we tested median or mean respirable PM mass concentrations in the corresponding two groups for each variable (as binary). In such analyses, we reported *p*-values from a Mann–Whitney test. All procedures were completed in NCSS 2021 (Utah, USA) and *p*-values below 0.05 were considered significant.

## 3. Results

We obtained data from 12 venues and completed 10 days of monitoring in each place. Monitoring duration ranged from four to twelve hours, depending on the open time in each place. Overall, we collected 11,831 data points, which were 5-min means of measurements, and the location of included venues is shown in Figure 1. Our study was performed in February, March, April, May, June and July 2022. We, therefore, could encompass both cold and warm seasons of the year. Table 1 offers basic information comparing venues with each other. We included venues with a wide range of internal floor area, from 24 to 550 m^3^, with the total staff ranging from 3 to 8 with up to 20 customers a day. Three included venues were located in former apartments, whereas two places were situated in a quite polluted neighborhood, close to a power plant. There were no windows in two salons out of twelve in the study. Table 1 also summarizes geometric mean and median respirable PM concentrations, calculated from ten daily means for each place.

Measured 5-min mean respirable PM concentrations ranged from 0 to 6.3 mg/m^3^ (median 0.073 (IQR 0.043; 0.157), geometric mean 0.083 mg/m^3^) with severe left skewness (8.40). Of the overall count of 11,831 data points, 5-min mean concentrations exceeding 0.1 mg/m^3^ were found in 4479 data points (38% of data), exceeding 0.2 mg/m^3^ in 2218 data points (19% of all data), and exceeding 0.5 mg/m^3^ in 614 points (5% of the data). Very high concentrations above 1 mg/m^3^ were recorded in 205 data points, which corresponded to 2% of the overall work time. All further analyses represent grouped daily data (122 days), either median or geometric means. Daily median respirable PM concentrations were highly variable and ranged from 0.013 to 0.666 mg/m^3^.

High daily median concentrations were recorded in the cold season only, whereas low daily concentrations were registered in June and July. We found high correlation between daily median (r = 0.83) respirable concentration inside the beauty salons with the mean daily ambient PM_2.5_ concentrations. Despite high variance of within-day concentrations, with spikes corresponding to the use of sprays and fans, the overall pattern of concentration changes from day to day followed that of the ambient PM_2.5_ concentration, suggesting that the latter was a strong predictor of the inside median concentrations. Nevertheless, the highest ambient daily concentration was 0.142 mg/m^3^, but the highest recorded indoor respirable PM concentration was as high as 6.3 mg/m^3^. This highest concentration was recorded in the salon number 12 during the intense and simultaneous use of sprays for more than one customer at a time

There was a highly significant difference between salons in the median indoor respirable PM concentrations (Kruskall–Wallis *p* < 0.001, power 100%) (Figure 2). The greatest within-salon variance was found in the venues #2 and #6. When we compared salons with each other, there was an 8.5-times difference in the median concentrations between the venue with the highest median (venue 2; 0.29 mg/m^3^) and the least median (venue 12; 0.034 mg/m^3^) of all daily median concentrations. In a multivariate linear regression modelling, accounting for the internal floor area, the mean number of customers a day, weekday (business day vs. weekend) and number of working personnel, the ambient PM_2.5_ concentration was the strongest and most powerful predictor of daily median respirable PM concentration in the salons (beta 2.12; 95% CI 1.89;2.39, *p* < 0.001, power 100%). This predictor alone explained 51% of the overall variability of the internal respirable PM concentrations (Figure 3), whereas the model with all abovementioned variables could explain 63% of the overall variability.

Salon staff were predominantly females (Table 2), college degree possessors (61%), of whom 37% were daily cigarette smokers, 29% smoked waterpipe regularly and 20% used electronic cigarettes on a daily basis. This population exhibited very low prevalence of respiratory symptoms and respiratory diagnoses, such as asthma or COPD. However, 25% mentioned they had seasonal allergy and 14% had chronic bronchitis. The scores of both HRQL domains in salon personnel were significantly higher compared to the general population of Almaty (Mann–Whitney *p* < 0.001). In the univariate analyses, testing the association of exposure to respirable PM at work, expressed as either the median of daily means for a given salon, with respiratory symptoms or diagnoses, most associations were non-significant, likely because of the small sample. We found positive association of the median respirable PM with the fourth question of CAT (shortness of breath) and seasonal allergy and negative association with chronic bronchitis (Table 3). The latter effect may indicate that hairdressers with confirmed diagnosis of chronic bronchitis avoid salons with high exposure levels. In addition, HRQL in the studied personnel was not associated with the mean levels of respirable PM in the workplace.

Daily smoking was significantly associated with respiratory symptoms (overall CAT score median 3 in daily smokers vs. 0 in non-smokers, *p* < 0.05; CAT question 1 (cough) median 1 vs. 0, *p* < 0.001; CAT question 2 (sputum) median 1 vs. 0, *p* < 0.01). Moreover, waterpipe use was significantly associated with a higher score on question CAT 2 (sputum). We did not find any associations of cigarette smoking with the diagnoses of chronic bronchitis, COPD, asthma or allergic rhinitis, likely because of the small number of cases.

## 4. Discussion

This study was planned to quantify exposure of beauty salon personnel to respirable PM, given that this occupational group is known to use a wide range of chemicals as sprays and liquids, many of which are allergens and may be associated with respiratory effects. We have now demonstrated that peak mass concentrations of respirable PM may be as high as 6.3 mg/m^3^, whereas daily median concentrations were highly variable and ranged from 0.013 to 0.666 mg/m^3^, also reflecting significant difference between the venues. Ambient fine PM concentrations were the strongest predictors of the indoor median respirable PM concentrations; the internal space or number of personnel working were not predictors. Overall, the prevalence of respiratory diagnoses in the studied population was low and was poorly associated with respirable indoor concentrations, most likely because of the dominance of young people and their as yet short work duration.

Occupational exposure to aerosols, especially respirable and fine fractions, is common and may explain many health effects in selected occupational groups. Such occupations and workplaces traditionally include metalworking [1,14], agriculture and construction [15,16], but also include occupations which earlier had little attention from public health advocates [17]. Despite ongoing debate over the exact size distribution of the respirable fraction (PM penetrating beyond ciliated airways) [18], exposure to respirable PM has been shown to have an association with many respiratory and cardiovascular outcomes and even cancer [19,20,21]. There is no clear understanding on the temporal distribution of industry-specific respirable PM resulting from frequent use of sprays in beauty industry, but given that the space is usually confined and the effect of ventilation is unknown, this occupational group will likely exhibit higher risks of respiratory and cardiovascular outcomes. However, the confounding effect of individual smoking and ambient air pollution hampers effect estimation.

One of the sound outcomes of our study was a powerful association of the median indoor respirable PM concentration in beauty salons with ambient fine PM. Furthermore, we found that air pollution in these workplaces in the cold season was much more pronounced compared to the warm season. We have earlier reported very high concentrations of PM10 in the open workplaces in Almaty [22], making all occupational groups working outdoors at risk of high exposure and the adjacent health effects. Such high levels of fine PM in Almaty result from the use of coal for heating, including the operation of the central power plant, providing hot water for this 2-million population megapolis. Our current study now further confirmed the contribution of heavy ambient air pollution in the exposure not only in those working outdoors, but also for those who stay inside for the entire workday. The issue of air pollution in Almaty and similar cities where suburbs widely use fossil fuel for heating during the cold season [23] is a clear call for public health action, necessitating urgent transition to the use of natural gas or alternative energy, such as nuclear generation. Our findings show that even people staying indoors in Almaty in winter are exposed to unhealthy levels of air pollution. Coupled with additive occupational exposure, as in our case, the risk may be fairly high.

The overall health status of beauty salon employees in our study was fairly good, given that their HRQL scores were even higher compared to the general population in the city, and respiratory symptoms were reported by very few subjects. We explain this with high prevalence of young people in their 20–30s in our sample, when chronic respiratory disease has not yet developed. On the other hand, cigarette smoking, use of waterpipe and electronic cigarettes use were reported quite often, and coupled with high levels of baseline air pollution and frequent use of sprays with a mix of chemicals in the workplace, this population will likely exhibit more respiratory and cardiovascular disease than the general population with longer-term employment. Such associations were identified in other countries [24,25,26,27]. Cases of asthma in hairdressers were linked to bleaching and the use of sprays, but most of these cases were reported as case series. In addition to respiratory effects, musculoskeletal disorders are often characterized as highly prevalent in hairdressers [24,25]; these are not related to inhalational exposures but to awkward posture.

Respirable PM median concentrations in these workplaces were low in comparison with the occupational exposure limits (OEL) used and enforced elsewhere. OELs for PM in Kazakhstan have never existed; therefore, compliance with regulations in this study cannot be assessed or documented. For workplaces, local legislation specifies total dust levels for selected workplaces including those with expected high crystalline silica content. Moreover, local legislation does not specify the difference between personal and area sampling. In the current study, we report area sampling, which is an alternative approach to test exposure, when personal sampling is not feasible.

The strength of our analysis is a large number of sampling measures and long sampling period, which included cold and warm seasons and was performed daily. However, the limitation is a small sample of workers, which resulted from poor consent to participate and general low interest in research. Another limitation is sampling of one PM fraction and no opportunity to monitor chemical composition of the PM arising from aerosols. In addition, we could not create a log of activities in the salons, which could associate the peaks of respirable PM with the use of sprays. Finally, respiratory presentation of the employees could not be verified for ancillary investigations, including spirometry, X-ray, because the overall compliance was quite poor.

## 5. Conclusions

In conclusion, this study allowed us to analyze 122 consecutive days of data, obtained from 12 beauty salons in Almaty reflecting real-time respirable PM mass concentrations. We have identified high variability of concentrations both within and between the salons. However, the median concentrations were more dependent on the ambient fine PM pollution, indicative of the enormous burden of ambient air pollution the cities such as Almaty. Although the respiratory health of included employees was quite fair and poorly associated with the median indoor respirable PM concentrations, more effort should be made to protect beauty salon staff from inhalable pollutants. We also call for urgent action for cigarette and waterpipe smoking cessation with the purpose of protecting respiratory and cardiovascular health.

## Figures and Tables

**Figure 1 ijerph-20-02429-f001:**
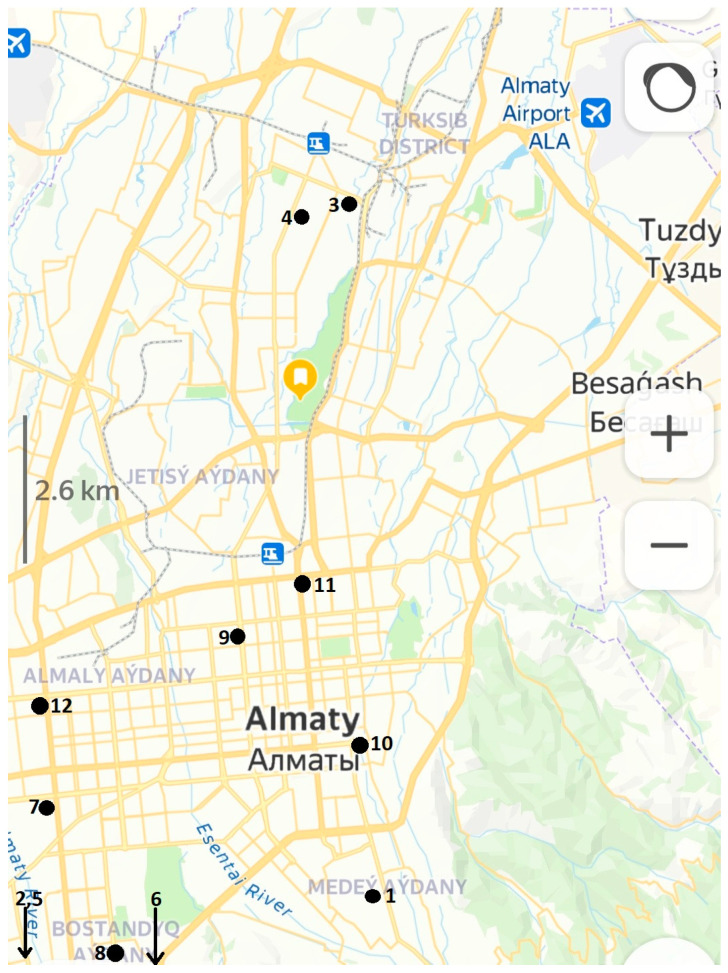
Location of included salons in Almaty, Kazakhstan (from yandex.kz).

**Figure 2 ijerph-20-02429-f002:**
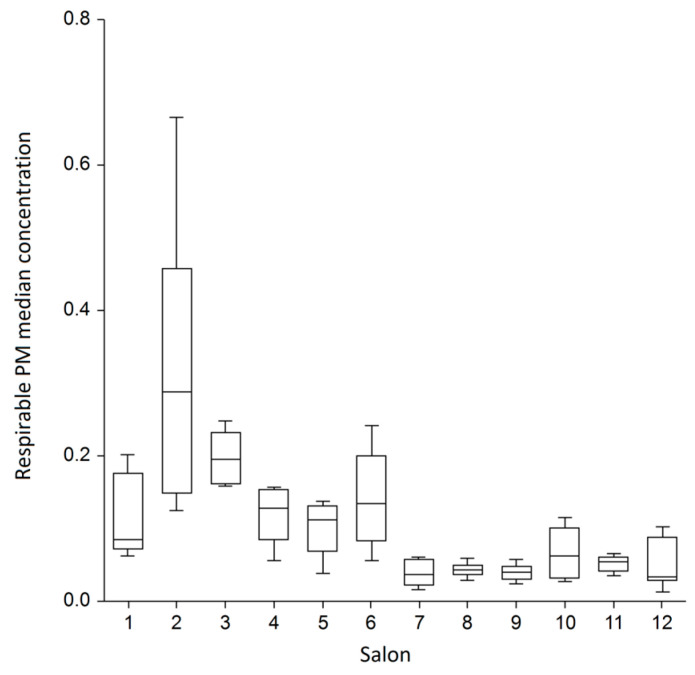
Daily median respirable PM concentrations (in mg/m^3^) in twelve sampled salons (Kruskall–Wallis *p* < 0.001). Boxes are interquartile ranges (IQR), whereas whiskers represent one more IQR.

**Figure 3 ijerph-20-02429-f003:**
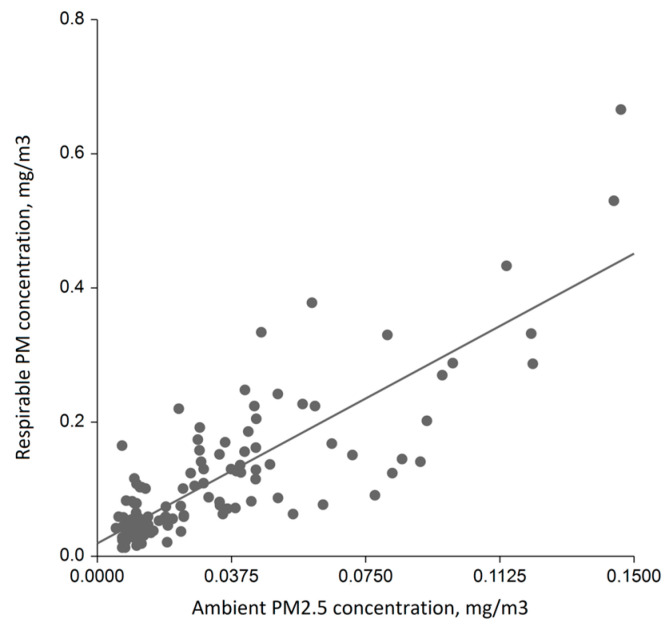
Scatter plot of respirable PM concentration in the indoor air of tested salons with regression line.

**Table 1 ijerph-20-02429-t001:** Summary table of included venues.

#	Area, m^3^	Staff	Customers a Day, Approx	Respirable PM, Geometric Mean of Daily Means, mg/m^3^	Respirable PM, Median of Daily Means, mg/m^3^	Days of Testing	Note
1	98	7	7–8	0.118	0.111	February	Large volume of sprays used for each customer, high prices
2	55	4	12–13	0.296	0.280	February	Former apartment
3	30	8	10–12	0.217	0.184	March	Located close to power plant
4	44	4	7–8	0.197	0.116	March	Located close to power plant
5	45	3	10–12	0.172	0.101	March	Former apartment
6	55	5	8–10	0.169	0.128	April	Former apartment
7	76	5	12–15	0.044	0.041	May	None
8	82	6	14–15	0.044	0.036	June	No windows
9	70	5	7–8	0.044	0.040	June	None
10	105	5	8–10	0.072	0.058	June	None
11	24	4	12–14	0.060	0.050	July	None
12	550	6	20	0.215	0.048	July	No windows

Note: PM—particulate matter.

**Table 2 ijerph-20-02429-t002:** Demographic and lifestyle profile of the salon personnel along with diagnoses and health-related quality of life.

Variable	
Females, N (%)	40 (78)
Age, mean ± SD, years	33.2 ± 9.3
Years in service, median (IQR)	6 (3–13)
BMI, mean ± SD, kg/m^2^	23.0 ± 2.6
Highest attained education, N (%)	
Secondary school	1 (2)
High school	9 (18)
College	31 (61)
University	10 (19)
Cigarette smoking, N (%)	
Never	24 (47)
Ex-smoking	5 (10)
Daily smoking	19 (37)
Occasional smoking	3 (6)
Waterpipe smoking, N (%)	15 (29)
Electronic cigarette use, N (%)	10 (20)
Exposure to secondhand smoke, N (%)	31 (61)
Cough (0–5), median (IQR)	1 (0–1)
Sputum (0–5), median (IQR)	0 (0–1)
Shortness of breath (0–5), median (IQR)	0 (0–0)
Wheezing, N (%)	3 (6)
Seasonal allergy, N (%)	13 (25)
Diagnosis of chronic bronchitis, N (%)	7 (14)
Diagnosis of COPD, N (%)	1 (2)
Diagnosis of asthma, N (%)	1 (2)
Diagnosis of allergic rhinitis, N (%)	3 (6)
PCS SF-8, median (IQR)	64.3 (62.3–67.3)
MCS SF-8, median (IQR)	65.4 (59.4–68.7)

Note: SD—standard deviation; IQR—interquartile range; COPD—chronic obstructive pulmonary disease; PCS—physical component score; MCS—mental component score.

**Table 3 ijerph-20-02429-t003:** Univariate comparisons of indoor median respirable PM concentrations (shown as mg/m^3^) for selected symptoms from the CAT questionnaire and diagnoses.

Variable	Yes	No
Cough (CAT, question 1, score ≥ 1)	0.05 (0.04–0.12)	0.06 (0.05–0.12)
Sputum (CAT, question 2, score ≥ 1)	0.05 (0.04–0.12)	0.11 (0.05–0.12)
Shortness of breath (CAT, question 4, score ≥ 1) *	0.12 (0.05–0.28)	0.05 (0.04–0.11)
Wheezing	0.13 (0.06–0.28)	0.05 (0.04–0.12)
Seasonal allergy *	0.12 (0.05–0.28)	0.05 (0.04–0.11)
Diagnosis of chronic bronchitis *	0.04 (0.04–0.05)	0.11 (0.05–0.13)
Diagnosis of COPD	One observation	0.06 (0.04–0.12)
Diagnosis of asthma	One observation	0.06 (0.04–0.12)
Diagnosis of allergic rhinitis	0.06 (0.04–0.28)	0.06 (0.04–0.12)

Note: *—<0.05 from Mann–Whitney U-test.

## Data Availability

The data presented in this study are available on request from the corresponding author.

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
