# Peer review of "Exposure to Respirable Particulate Matter and Its Association with Respiratory Outcomes in Beauty Salon Personnel"

_ijerph, 2023, doi:10.3390/ijerph20032429_

Round 1

Reviewer 1 Report

The paper presented an interesting study and clearly written.  I do not have significant suggestions and they are provided below:

1.       It would benefit from some minor editing to improve the English.  I will list a few examples, but these are not major concerns:

Line 37, add “the” before occupations.

Line 48, add “to” before change

Line 51 – would suggest using “mixture” and not “mix”, same comment in line 61

Lines 52-55 – suggest this section be rewritten for ease of reading.

Line 73 – with “the” Declaration

Line 80 – suggest changing “10-days periods” to “10-day periods”

Line 84 – suggest changing “fill in” to “to complete” a questionnaire

Line 85 – suggest inserting “or” in place of “of” between Russian and Kazakh

Lines 93-94 – suggest deleting “ever” in phrase “ever-doctor”

Lines 98-99 – suggest rewording “late until the last customer” to something like “late into the day” or similar

I did not continue with editorial comments and this type of editing is suggested throughout the manuscript. Again, these are not major issues but such changes would improve the paper.

2.       Line 101 – it is suggested that the detection method used by the TSI AM550 be stated (I believe it is light scattering photometric but this should be checked). 

3.       Line 125 states that the data are not normally distributed, does this also man that the data are not log-normally distributed.  With air samples (especially ones with a wide range of distribution), log transformed data is commonly used. Later in the paper geometric mean values are presented along with mean values.  I found this somewhat confusing.  If the distribution is log normal, the central tendency of the data should best be represented by the GM.  This aspect needs to be further explained.

4.       All the mean values were very low in comparison to allowable occupational exposure values for PM.  Given that the study was intended to assess the employee exposure within beauty parlors, I suggest that the conclusion section provide some discussion of this point.  The relationship to ambient PM2.5 values is interesting, but it seems as though the primarily discussion pertained to it without any real discussion of occupational exposure values.  This conclusion would need to be qualified in that all the samples were area values and not personal values.

5.       Line 188 reports that one sample had a value of 6.3 mg/m3.  It would be interesting to learn more about this value and any speculation as to why it was so much higher than any of the means.  Did it come from the site with the highest mean value?  Was it associated with any specific employee activity?

Author Response

The paper presented an interesting study and clearly written.  I do not have significant suggestions and they are provided below:

We thank the Reviewer for a deep appraisal of our manuscript along with the valuable comments we have attempted to address. Below we offer a point-by-point response to the issues raised.

  1. It would benefit from some minor editing to improve the English.  I will list a few examples, but these are not major concerns:

Line 37, add “the” before occupations.

Line 48, add “to” before change

Line 51 – would suggest using “mixture” and not “mix”, same comment in line 61

Lines 52-55 – suggest this section be rewritten for ease of reading.

Line 73 – with “the” Declaration

Line 80 – suggest changing “10-days periods” to “10-day periods”

Line 84 – suggest changing “fill in” to “to complete” a questionnaire

Line 85 – suggest inserting “or” in place of “of” between Russian and Kazakh

Lines 93-94 – suggest deleting “ever” in phrase “ever-doctor”

Lines 98-99 – suggest rewording “late until the last customer” to something like “late into the day” or similar

I did not continue with editorial comments and this type of editing is suggested throughout the manuscript. Again, these are not major issues but such changes would improve the paper.

We indeed thank the Reviewer for this valuable comment. We have now introduced all proposed edits and further checked the manuscript for spelling and especially for the use of articles. We hope that the readability of this manuscript has further improved with such proofreading.

  1. Line 101 – it is suggested that the detection method used by the TSI AM550 be stated (I believe it is light scattering photometric but this should be checked). 

Correct, this device uses lase scattering photometric technology. We have placed an explanation on that in the text.

  1. Line 125 states that the data are not normally distributed, does this also man that the data are not log-normally distributed.  With air samples (especially ones with a wide range of distribution), log transformed data is commonly used. Later in the paper geometric mean values are presented along with mean values.  I found this somewhat confusing.  If the distribution is log normal, the central tendency of the data should best be represented by the GM.  This aspect needs to be further explained.

This is indeed true that the data were non-normally distributed, and another Reviewer has also addressed the issue of data normality and proposed to report beta coefficients from the regression models, in which we now analyzed log-transformed median indoor concentrations as a dependent variable and log-transformed ambient concentrations as a predictor. We thank the Reviewer for this very pertinent comment. Moreover, we understand that mean concentrations should not normally be reported. However, we intentionally reported that because in two previous studies we have published the Reviewer insisted on demonstrating mean concentrations along with the major analysis of the medians for the illustrative purposes, which we considered odd. Nevertheless, we included those concentrations there too. Given that conceptually we agree that in case of non-normal data, reporting mean concentrations is not a conventional practice and that the Reviewer confirms now such approach, we have now deleted statements that we report mean concentrations too from the Methods and Results. We hope that the current presentation is less confusing and is consistent with data normality analysis outcomes. 

  1. All the mean values were very low in comparison to allowable occupational exposure values for PM.  Given that the study was intended to assess the employee exposure within beauty parlors, I suggest that the conclusion section provide some discussion of this point.  The relationship to ambient PM2.5 values is interesting, but it seems as though the primarily discussion pertained to it without any real discussion of occupational exposure values.  This conclusion would need to be qualified in that all the samples were area values and not personal values.

This is indeed true that the median concentrations may have been blow OELs used elsewhere, but OEL for PM of any diameter do not exist in Kazakhstan. We do consider the comment very important and, therefore, now added a narration on that in the Discussion to read: “Respirable PM median concentrations in these workplaces were low in comparison with the occupational exposure limits (OEL) used and enforced elsewhere. OELs for PM in Kazakhstan have never existed; therefore, compliance with regulations in this study cannot be assessed or documented. For workplaces, local legislation specifies total dust levels for selected workplaces including those with expected high crystalline silica content. Moreover, local legislation does not specify the difference between personal and area sampling. In the current study, we report area sampling, which is an alternative approach to test exposure, when personal sampling is not feasible.”

  1. Line 188 reports that one sample had a value of 6.3 mg/m3.  It would be interesting to learn more about this value and any speculation as to why it was so much higher than any of the means.  Did it come from the site with the highest mean value?  Was it associated with any specific employee activity?

This very high concentration was surprising to us too, and in fact came from the salon 12 with the least median concentration. We now went back to the activity log and found that this concentration lasted for a very short time (10 min) and coincided with a very intense use of sprays by two or more hairdressers at a time, since this was the venue with the largest personnel. Given that this is worth mentioning, we have added a clarification in the corresponding line in the Results: “This highest concentration was recorded in the salon 12 during the intense and simultaneous use of sprays for more than one customer at a time”.

Once again, thank you for this detailed review. We believe that the manuscript is substantively improved with these changes. 

Sincerely,

Dr Denis Vinnikov, Corresponding Author

Reviewer 2 Report

In this paper "Exposure to respirable particulate matter and its association with respiratory outcomes in the beauty salon personnel" the authors aimed to assess exposure to indoor particulate matter.

They achieve this also using suitable measuring instruments.

The paper describe the methodology and it does clear effort by the authors that submitted also the questionair to the workers.

However, I propose some suggests that could usefull to improve work.

1.    I suggest to insert a map of geocoded salons.

2.    How many people refuse partecipation? Are they different by gender, age and other socio-economic carachteristics from the partecipant? could be a selection bias?

3.    The sentence " In total, 11831 data points were analyzed, which corresponded to 122 days of monitoring, 10 days from each of 12 included salons" in the section method is also reported in the section results, so I suggest to eliminate it.

4.    because the authors claim that the daily concentration of PM is not normally distributed, in the regressione model to identify predictor of daily indoor respirable PM concentration they should insert log-trsformated of PM as dipendent variable. The exponent of beta coefficient will rappresent a risk measure  called Geometric Mean Ratio.

5.     The authors write "We then aimed to see whether indoor respirable PM concentrations were associated with 150 respiratory symptoms, respiratory ever-diagnoses or HRQL" , howevere they did not performed any statistical models but they tested only the differences among group and they cannot draw any conclusions about the association.

6.     Could be usefull for the reader to have a table 3 with the results of the tests that the authors describe in the text.

Author Response

In this paper "Exposure to respirable particulate matter and its association with respiratory outcomes in the beauty salon personnel" the authors aimed to assess exposure to indoor particulate matter.

They achieve this also using suitable measuring instruments.

The paper describe the methodology and it does clear effort by the authors that submitted also the questionair to the workers.

However, I propose some suggests that could usefull to improve work.

We thank the Reviewer for a deep appraisal of our manuscript along with the valuable comments we have attempted to address. Below we offer a point-by-point response to the issues raised.

  1. I suggest to insert a map of geocoded salons.

We thank the Reviewer for this suggestion and agree that the map would help a reader better understand our results and conclusions. We have placed such map as Figure 1 and renumbered all subsequent figures.

  1. How many people refuse partecipation? Are they different by gender, age and other socio-economic carachteristics from the partecipant? could be a selection bias?

Thirty hairdressers refused to participate. We could not assess the socioeconomic characteristics of those who refused to participate, but sex and age did not statically differ from those who agreed, although there was a trend of more females in the groups of those who refused. We have added an explanation on that in the Methods to read: “We found not find differences in age and sex between the groups of those who agreed and refused to participate.”.

  1. The sentence " In total, 11831 data points were analyzed, which corresponded to 122 days of monitoring, 10 days from each of 12 included salons" in the section method is also reported in the section results, so I suggest to eliminate it.

Done. Thank you.

  1. because the authors claim that the daily concentration of PM is not normally distributed, in the regressione model to identify predictor of daily indoor respirable PM concentration they should insert log-trsformated of PM as dipendent variable. The exponent of beta coefficient will rappresent a risk measure  called Geometric Mean Ratio.

      We indeed thank the Reviewer for this pertinent comment. We also acknowledge that we missed this issue of log transformation of data for the analysis at the preceding stage. Given that this issue was also addressed by another reviewer, we have rerun the regression analysis, which now considers log-transformed ambient concentrations as an independent variable and log-transformed median indoor concentration as a dependent variable, accounting for confounders. Corresponding edits made in the Methods, Results and the Abstract. We demonstrate now that the effect has changed to beta of 2.12, whereas the corresponding R2 for the whole model and R2 for the model when the principal predictor (ambient concentration) has also changed.

      Because another reviewer has also raised the issue of the usefulness of the mean concentrations, we have now deleted the sentence elucidating the effect in the regression model with the mean indoor concentrations as a dependent variable.

  1. The authors write "We then aimed to see whether indoor respirable PM concentrations were associated with respiratory symptoms, respiratory ever-diagnoses or HRQL" , howevere they did not performed any statistical models but they tested only the differences among group and they cannot draw any conclusions about the association.

This is true, we only reported univariate comparisons for this analysis. Given that this clarification is pertinent, we have elected to clarify the test we used in the cited sentence, which now reads: “We then aimed to see whether indoor respirable PM concentrations differed in the univariate comparisons of respiratory symptoms, respiratory diagnoses or HRQL; for that we tested median or mean respirable PM mass concentrations in the corresponding two groups for each variable (as binary)”.

  1. Could be usefull for the reader to have a table 3 with the results of the tests that the authors describe in the text.

We have now added Table three with the univariate comparisons, as advised. The corresponding test in the Results, Discussion and even Abstract was slightly amended to conform with the findings.

Once again, thank you for this detailed review. We believe that the manuscript is substantively improved with these changes. 

Sincerely,

Dr Denis Vinnikov, Corresponding Author